# Seroprevalence of *Brucella* Infection in Wild Boars (*Sus scrofa*) of Bavaria, Germany, 2019 to 2021 and Associated Genome Analysis of Five *B. suis* Biovar 2 Isolates

**DOI:** 10.3390/microorganisms11020478

**Published:** 2023-02-14

**Authors:** Laura Macías Luaces, Kerstin Boll, Corinna Klose, Janina Domogalla-Urbansky, Matthias Müller, David Eisenberger, Julia M. Riehm

**Affiliations:** 1Bavarian Health and Food Safety Authority (LGL), Department of Animal Health, 85764 Oberschleißheim, Germany; 2Bavarian Health and Food Safety Authority (LGL), Department of Animal Health, 91058 Erlangen, Germany

**Keywords:** brucellosis, *Brucella suis* biovar 2, monitoring, wild boar, ELISA, CFT, PCR, genome sequencing

## Abstract

*Brucella* species are highly pathogenic zoonotic agents and are found in vertebrates all over the world. To date, Germany is officially declared free from brucellosis and continuous surveillance is currently limited to farm ruminants. However, porcine brucellosis, mostly caused by *B. suis* biovar 2, is still found in wild boars and hares. In the present study, a three-year monitoring program was conducted focusing on the wild boar population in the state of Bavaria. Serologic screening of 11,956 animals and a direct pathogen detection approach, including a subset of 681 tissue samples, was carried out. The serologic incidence was 17.9%, which is in approximate accordance with previously published results from various European countries. Applying comparative whole genome analysis, five isolated *B. suis* biovar 2 strains from Bavaria could be assigned to three known European genetic lineages. One isolate was closely related to another strain recovered in Germany in 2006. Concluding, porcine brucellosis is endemic in Bavaria and the wild boar population represents a reservoir for genetically distinct *B. suis* biovar 2 strains. However, the transmission risk of swine brucellosis to humans and farm animals is still regarded as minor due to low zoonotic potential, awareness, and biosafety measures.

## 1. Introduction

Various species of the genus *Brucella* are highly pathogenic zoonotic agents and may cause severe infections in humans and vertebrates, mostly mammals. Although brucellosis is known since almost 200 years from now, it was identified as cause of death in reptiles, fish, and frogs only in the last decades [1]. The most known, *B. melitensis*, *B. abortus*, and *B. suis* are broadly spread worldwide and animal brucellosis has a significant economic impact [2]. The agent was transmitted via milk, food products, or during handling infected individuals. Until the 1950s, the disease was highly endemic in humans and farm animals in Germany [3]. Human brucellosis cases declined in the late 1970s and to date most of these result from infections abroad [4,5]. Due to strict and expensive eradication programs in the last decades, the farm animal population is currently designated as brucellosis-free in Germany [6]. However, wild-living animals are still considered possible natural reservoirs, among these boars, hares, and voles. Data from northeastern Germany investigating 763 wild boar serum samples from 1995 to 1996 revealed a seroprevalence of 22.0% [3]. In the federal state of Saxony, 31,417 serum samples were investigated and showed an average positive rate of 20.7% [7]. On the contrary, the seroprevalence in the federal state of Baden Württemberg in 2018 revealed only three reactive out of 1224 samples (0.2%) [8]. Published studies from Europe, Finland, Latvia, and the Czech Republic report comparably low seroprevalences, with 9.2%, 9.6%, and 8.7% [9,10,11]. Whereas in Southern Europe, Croatia, and some regions of Spain, rather high seroprevalences were published, with 27.6%, 29.7%, and even 59.3%, respectively [11,12,13]. As well, cultural diagnostics of *Brucella* was successful in some of the wild boar samples, and mostly isolates of B. suis biovar 2 were recovered [14,15].

In the last decades, molecular identification and typing methods, as well as genome sequencing of *B. suis* biovar 2 isolated from domestic pigs and wildlife revealed diverse genetic lineages of the pathogen in Europe [1]. Interpretation of results suggested individual geographically focused genotypes originating especially from smaller mammals, such as hares. As well, comparably broad distributed lineages spread over long distances that were recovered from wildlife hosting large territories, such as wild boar [16,17].

To determine the serologic incidence of brucellosis in the wild boar population, a three-year monitoring program was conducted and evaluated in the present study. Molecular investigation and comparative whole genome analysis were scheduled to reveal possible lineages of *B. suis* biovar 2 in Bavaria, southern Germany.

## 2. Materials and Methods

Within the framework of the present study, volunteer Bavarian hunters carried out the sampling of the wild boar population during the official hunting seasons from 2019 to 2021. During this time, a financial reimbursement was paid for each shot animal in Bavaria due to the high density of wild boars and the impending African swine fever pandemic, irrespective of the present study [18]. However, animals were hunted primarily for their meat, preservation of the countryside, and finally due to the lack of natural predators. In Germany, hunters complete a comprehensive training before they are allowed to hunt down wild vertebrate animals. The rangers routinely eviscerated the dead animals after shooting, inspected the carcasses for diseases or prior injuries, and then took samples for the subsequent laboratory investigation. There was no standardized health condition statement reported due to an unacceptable effort and limited conveyance possibilities. If a wild boar was found dead, the entire carcass was directly delivered to our pathology department for necropsy. From all individuals, a serum or transudate sample was collected from the visceral cavity into serum tubes. Additional tissue samples were randomly collected from a smaller subset of animals and were cut out with a clean knife according to the capacity of the individual hunter. In some rare cases, organ samples were collected from animals showing macroscopic abnormalities. Samples were either directly transferred to the Bavarian Health and Food Safety Authority or stored until shipment at 4 °C.

### 2.1. Serologic Examination

The initial serologic screening was carried out using a commercially available enzyme-linked immunosorbent assay (ELISA) ID Screen Brucellosis Serum Indirect Multi-Species Kit (ID Vet, Grabels, France), according to the manufacturer’s instructions. The indirect ELISA was developed for the detection of antibodies against *B. abortus, B. melitensis*, and *B. suis*. As suitable specimens, serum or plasma may be used. The test kit was validated to ensure sensitive detection as well in pool samples. Repeat testing was performed with non-negative samples. Additionally, non-negative samples were heat inactivated at 58–60 °C for 30 min and centrifuged at 2000× *g* for 5 min before retesting using the complement fixation test (CFT) according to the standard procedure published by the World Organization for Animal Health (WOAH) [19,20]. A reactive result was assigned if the CFT revealed ≥20 sensitive units/mL. If the result of the CFT was invalid and the serum revealed unacceptable quality, the respective sample was excluded from the study. All serum samples with a reactive result were preserved at −80 °C for future studies.

### 2.2. Pathological Investigation

Carcasses were subject to necropsy and a macroscopic post-mortem examination. Approximate age and sex were determined, as well as prior injuries and noted within the individual files. Selected organ samples, such as the testicles, uterus, lymph nodes, and spleen tissues, were fixed using a 4% formaldehyde solution and embedded in paraffin wax according to standard procedures. The tissue slices underwent HE coloring and were examined microscopically. Tissue samples of each investigated animal were subject to a bacteriological and molecular investigation targeting *Brucella*. All preparations and microscopic slides with a positive outcome were preserved for future studies.

### 2.3. Bacteriological Analysis

For the bacteriological investigation, tissue samples of wild boars, including testicles, uterus, lymph nodes, and spleen were dissected under sterile conditions and plated on selective *Brucella* agar (Oxoid, Wesel, Germany). Besides the nutritional ingredients, this medium contains a supplement cocktail. Due to the slow growth of *Brucella* species, effective suppression of other bacteria and fungi needs to be achieved during incubation. The substances polymyxin B, bacitracin, nalidixic acid, and vancomycin are summarized as antimicrobials. The substance nystatin has an antifungal effect. Cultural incubation was performed in a microaerophilic atmosphere at 37 °C and 10% CO_2_ for five to seven days [19,20]. To meet biosafety standards, suspicious *Brucella* colonies were transferred into a BSL-3 laboratory, confirmed by MALDI-TOF mass spectrometry (Bruker, Bremen, Germany), and preserved at −80 °C until further investigation.

### 2.4. Molecular and Genome Investigation

All tissue samples were screened by real-time PCR. DNA isolation was carried out using the DNeasy Blood & Tissue Kit according to the manufacturer’s instructions (Qiagen, Hilden, Germany). Besides other material, the test kit is optimized for fresh or frozen animal tissue and bacteria. Pieces of 25 mg of tissue sample were initially incubated with proteinase K for 20 h to achieve the complete lysis. Regarding the total amount of 10^11^ bacterial cells, the incubation time was limited to 6 h in the present study. For the molecular screening of tissue DNA, the specific *Brucella* Cell Surface Protein 31 (*bcsp31*) gene was targeted according to a previously published protocol [21]. As an internal amplification control we used the IC2 template, primer and probe set to prevent false negative PCR results due to inhibition for all samples.

Regarding whole genome sequencing of *Brucella* isolates, DNA extraction was carried out using as well the DNeasy Blood & Tissue Kit according to the manufacturer’s instructions (Qiagen, Hilden, Germany). The purified DNA was quantified and adjusted using a Qubit dsDNA HS Assay Kit (Life Technologies, Waltham, MA, USA) using a Qubit 4.0 fluorometer (Invitrogen, Carlsbad, CA, USA). Library preparation was performed using a Nextera XT DNA Library Prep Kit and a Nextera XT Index Kit according to the manufacturer’s instructions (Illumina, San Diego, CA, USA). Libraries were quantified using an Agilent High Sensitivity DNA Kit (Agilent Technologies, Waldbronn, Germany) on a 2100 Bioanalyzer Instrument (Agilent Technologies, Santa Clara, CA, USA). On an Illumina MiniSeq system, 2 × 150 bp paired-end reads were generated. Sequenced reads with a mean assembly coverage depth of 134 × (range 107 to 172) were analyzed.

Verification of the species *B. suis* and the biovar 2 identification out of the five biovars as well as the closely related species *B. canis* was carried out by in silico PCR according to the previously published Suis-ladder and using Mash distance implemented in SeqSphere+ software version 7.3.4 (Ridom GmbH, Münster, Germany) [22]. A core genome multilocus sequence typing (cgMLST) scheme developed for *Brucella* species and validated on 612 genomes was applied using the SeqSphere+ software [23,24]. Alignment and neighbor-joining investigation was carried out, including 61 isolates originating from the countries of Belgium, Bulgaria, Czech Republic, Denmark, France, Germany, Hungary, Italy, Portugal, Romania, Spain, and Switzerland, accessed through the National Center for Biotechnology Information (NCBI) database. We limited the included sequences to the isolation origin from European countries due to the geographic range. For the visualization of clonal relationships regarding the alleles, minimum spanning trees (MSTs) were created in SeqSphere+ using the setting “pairwise ignoring missing values”. Closely related isolates with a maximum difference of three alleles were subsequently assigned to clusters. As previously published this technology has become widely accepted for outbreak investigation, tracing and genomic epidemiology [24,25]. The raw sequencing data were published at the European Nucleotide Archive (ENA), project number PRJEB55696.

## 3. Results

The present study on brucellosis in wild boars revealed a serologic incidence of 17.9% in Bavaria, Germany. Direct pathogen detection by PCR was successful in eight cases (1.2%), and genome analysis was conducted on five *B. suis* biovar 2 isolates.

### 3.1. Sample Composition Regarding the Wild Boar Population in Bavaria, Germany

From January 2019 to December 2021, samples of 11956 wild boars were serologically investigated at the department of pathology and bacteriology of the Bavarian Health and Food Safety Authority in southern Germany. From a subset of 681 individuals, additional tissue samples, including testicles, uterus, lymph nodes, and spleen tissues, were analyzed in a direct pathogen detection approach (Table 1). Due to higher handling and shipping efforts, hunters did not provide more tissue samples, especially from animals that were considered and evaluated for consumption.

### 3.2. Serologic Incidence

During the three-year study period, serum samples from 11,956 wild boars were analyzed. The share of reactive samples varied between 17.6% and 18.2% during the investigation period, and the average was 17.9% (Table 1). The geographic distribution regarding the incidence of *Brucella*-reactive serum samples varied between the administrative districts of Bavaria. The samples originated from all seven administrative districts. Most of the samples were collected in the geographically largest district of Upper Bavaria, *n* = 3395, and 405 out of these were reactive (11.9%, Figure 1). Fewer samples were provided from Upper Franconia, *n* = 2159, with 351 reactive samples (16.3%); from Upper Palatinate, *n* = 1968, with 375 reactive (19.1%); from Lower Bavaria, *n* = 1576, with 426 reactive (27.0%); from Lower Franconia, *n* = 1458, with 197 reactive (13.5%); from Swabia, *n* = 824, with 207 reactive (25.1%); and finally from Middle Franconia, *n* = 576, with 180 reactive samples, respectively (31.2%, Figure 1).

### 3.3. Pathology Results

In all, 18 carcasses of wild boars were subject to pathologic examination by veterinarians regarding brucellosis during the investigation period. According to the degree of the decay, age and sex were determined. The exterior was inspected and prior injuries were described. Emaciation, parasitic infestation and recent wounds were considered as well as signs for infectious diseases. In three out of eight PCR-positive cases, suspicious findings regarding a *Brucella* infection were also marked in necropsy. Two animals showed specific macroscopic lesions in one or both testicles. Due to inflammation and necrosis of the testis and the adjacent epididymis, large amounts of purulent exudate accumulated in the scrotum (Figure 2). The third animal exhibited a chronic, purulent inflammation (abscessation) of the cervical and inguinal lymph nodes (Table 2). Organ samples of these individuals microscopically revealed granulomatous infiltration of lymphatic structures. Furthermore, associated lymph nodes were massively augmented and contained large numbers of macrophages.

### 3.4. Bacteriological and Molecular Investigations

The cultural investigation of tissue samples originating from 681 wild boars yielded five *Brucella* isolates (0.7%) (Table 1). As confirmed by the National Reference Laboratory for animal brucellosis in Jena, Germany, the isolates were identified exclusively as *B. suis* biovar 2. The strains were transferred into a BSL-3 laboratory and preserved at −80 °C for further investigation.

The molecular screening for the *Brucella* specific *bcsp31* gene regarding these samples revealed eight positive individuals (1.2%), three more than in cultural analysis (Figure 1 and Table 2). The three cultural negative animals were found dead, two out of these presumably for more than three days and explaining the unsuccessful cultural investigation.

### 3.5. Whole Genome Investigation of Five Brucella suis Biovar 2

Genome sequencing and genome analysis was successful for all collected strains. Four isolates revealed the *Brucella* specific cgMLST sequence type (ST) 16 and one isolate ST15 (Table 2). Consequently, no new genetic lineages were identified within the present study. Isolates 163 and 168 were assigned to the same cluster and complex type 3 with an allelic distance of three. The respective wild boar cadavers were found approximately 60 km apart. Isolate 162 revealed six alleles distance to the cluster of isolates 163 and 168, while the two remaining isolates showed a distance of more than 100 alleles (Figure 3). Among the further 61 European sequences, isolate 162 clustered with two German isolates from wild boars found in 2006, GCA_000371245.1 and GCA_000371225.1 (Appendix A and Figure 4). However, no detailed information regarding their specific geographic origin was available in the NCBI database.

## 4. Discussion

In the scope of the present study, 11,956 animals were screened for brucellosis in Bavaria, Germany. The number of investigated porcine samples varied regarding their geographic origin due to the diverging density of wildlife in the seven districts of Bavaria. Although these animals prefer forested habitats, they adapt quickly and occupy terrains close to human settlements [26]. Consequently, 44.9% of the samples originated from Upper Bavaria and Upper Palatinate. Furthermore, the seroprevalence was not found uniformly distributed and ranged from 11.9% in Upper Bavaria to 31.2% in Middle Franconia (Figure 1). The latter represents the district, where the report of one clinically manifest brucellosis case in a wild boar became known among the local hunters (Table 2 and Figure 2). As consequence, a comparably high number of samples was sent in for analysis. The results revealed an elevated prevalence and suggested a geographically focused endemic hotspot (Figure 1). Another reason for the varying numbers of tested animals was the monitoring-based nature of the study. Voluntary sampling was conducted instead of a precalculated and scheduled acquisition of wild boar specimens. Hunters decimate the number of wild boars due to variable reasons, including trade of game, landscape protection, control of infectious diseases or parasites, and finally compensation of the lack of predators [26]. Regarding the brucellosis monitoring on wild boar, representative and statistically evaluable results on a district-based calculation in Bavaria were nonetheless assured due to the large amount of investigated specimens (Table 1 and Figure 1). Investigation of incidences on wildlife animals may lead to varying results due to differences in the study design. Previously published reports range from 0.2% to 22.0% in Germany, and from 8.2% to even 59.3% in other regions in Europe, and support this assumption [2,3,7,8,9,10,11,12,13]. As well, various disease situation events on brucellosis in livestock were monitored by the World Organization for Animal Health (WOAH) [27]. The overall incidences indicate a broad endemic presence of the pathogen in natural foci with wild boars as a reservoir host in Central Europe.

The available diagnostic tests for indirect pathogen detection, CFT, and ELISA, are listed as standard methods in the diagnostic brucellosis manual released by the WOAH [19]. As published, these serology tests provide sensitivity rates from 89% up to 94%. Furthermore, the specificity in *Brucella* serology was determined at only 85% for ELISA and 96% for CFT, and this is considered comparably low [28]. Various cross-reactive parameters are assumed, including antibodies against *Salmonella*, *Yersinia*, *Escherichia*, *Francisella*, *Moraxella,* or further antigens that are prevalent in natural habitats [19,28,29]. The determined positive rates in published studies, as well as in the present study, might therefore differ from the true brucellosis seroprevalence. In order to reduce the rate of false positive results in the present study, a sample was only defined as reactive if both tests, CFT and ELISA, were positive. However, the CFT requires high-quality sample material. As known, serum samples originating from wild animals are more difficult to collect and transferred; these may become hemolytic easily, and contain a broader variety of antibodies [19]. Therefore, some of the collected specimens were excluded from the present study. It may therefore be concluded that the prevalence rates in the present study, as well as in previously published studies, do not necessarily reflect the number of true seroconversions due to brucellosis. Again, this gray area has been addressed in the past [19].

The applied direct pathogen detection approaches for brucellosis, including the cultural and molecular analysis, are sensitive and highly specific [19,20]. However, antigen detection is rather successful when individuals undergo an acute *Brucella* infection and is, therefore, temporarily limited. It may also be successful if *Brucella* persists in the investigated organs showing pathologic lesions. In the present study, most samples originated from clinically inapparent animals, and the molecular detection rate was 1.2%. The cultural investigation was even less successful than the molecular approach by PCR (Table 1). In Germany, the previously published numbers ranged from 1.4% to 2.9% [7,30]. The serologic incidence, on the other hand, revealed 17.9% in the present study. If the sero-reactive but PCR-negative animals ever underwent a previous *Brucella* infection, it must remain unanswered at this point.

The molecular characterization and typing of bacteria has been carried out for only a few decades. In 2007, multi locus sequencing analysis (MLSA) was published to amend the biotyping of *Brucella* [31]. The multi locus variable number of tandem repeat analysis (MLVA) and further genome analysis completed epidemiological assumptions and evidence of phylogenetic evolution regarding *Brucella* species [1,15,16,17]. *Brucella* genomes have been analyzed now for two decades, so far revealing that *Brucella* species are highly related to each other. Only few years ago, new *Brucella* species were identified in non-mammalian vertebrates, characterized and fill up evolutionary niches [1]. *B. suis* is known since more than one century now. The molecular clustering of the five biovars of *B. suis* also includes *B. canis* [32,33]. Studies on the epidemiology of this pathogen using molecular MLVA on 68 *B. suis* biovar 2 from Europe showed that various isolates from hares reveal individual lineages in a limited geographic range. An explanation is that these animals live in a bounded habitat. On the contrary, the wild boar lives within a rather extensive natural range, shows high adaptability to diverse habitats, and is named an invasive species [26]. Consequently, the molecular typing of *B. suis* biovar 2 originating from wild boar revealed a broad variety of lineages [16]. This finding was confirmed in the present study with three different genetic lineages among the five isolated strains. At the same time, the close genetic relationship of isolate 162 recovered in 2019 with two isolates from 2006 in Germany prove that *B. suis* biovar 2 has been endemic at least for 13 years (Figure 4 and Appendix A). This underlines that Bavaria in endemic for *B. suis* biovar 2.

Until the early 1980s, human brucellosis was endemic in Germany. At this time, *B. melitensis* was the most frequently identified *Brucella* species [4]. The animal reservoir consisted mostly of sheep or goats [3]. National eradication programs were strictly followed to eliminate seropositive animals since then. As recently published in Belgium, cattle were infected with *B. suis* biovar 2, transmitted from pigs [27]. Currently, European law considers porcine brucellosis as a category E disease and demands the need only for surveillance [34]. However, *B. suis* has been shown to be transmissible from wild boars to semi-free living domestic pigs. In 2018, a brucellosis outbreak was notified in the north of Germany on an outdoor pig farm [35]. The national monitoring program does not include domestic pigs due to high costs, the shorter overall life span in pigs, the low pathogenicity of porcine brucellosis for humans, and finally mandatory biosecurity measures in agricultural pig farms [36]. Simultaneously, the transmission risk from wild boars to farm animals, especially swine, is estimated moderate or low due to limited interactions and lack of sexual contact [37]. However, due to endemic occurrence, porcine brucellosis should not be underestimated.

## 5. Conclusions

As published in several studies from Europe, an average serologic incidence of one-fifth and the sporadic detection of *Brucella suis* biovar 2 regarding the wild boar population was found as well in Bavaria, Germany. A geographically broad distribution and the different genetic lineages of isolated strains suggest that the pathogen is endemic. Currently, the risk of transmission of swine brucellosis to humans and farm animals is still regarded as minor due to low zoonotic potential, awareness, and biosafety measures. However, it must not be underestimated and should be monitored regularly.

## Figures and Tables

**Figure 1 microorganisms-11-00478-f001:**
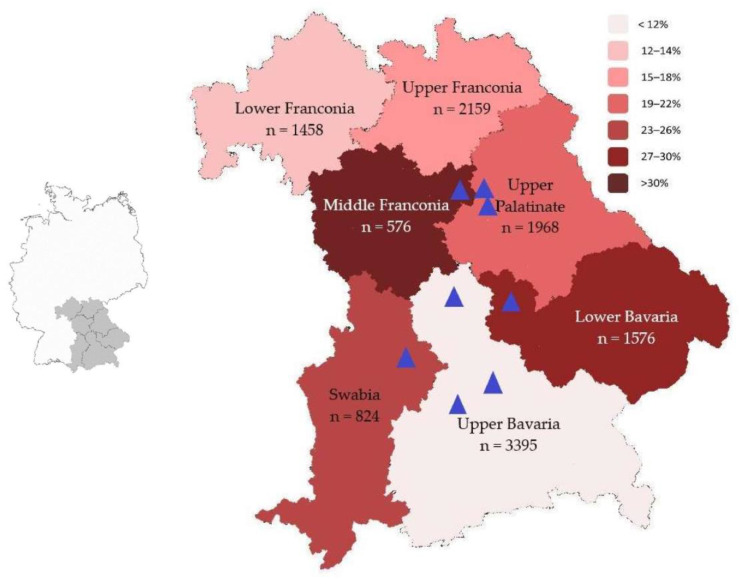
Map of Germany with Bavaria shaded in grey (**left**); map of Bavaria (**right**) showing serologic incidence and geographic distribution regarding brucellosis in wild boars from 2019 to 2021. Incidences are represented in a color scale from low (**light**) to high (**dark**). The total number of examined individuals (n) is shown for each district. The blue triangles represent direct pathogen detection by PCR.

**Figure 2 microorganisms-11-00478-f002:**
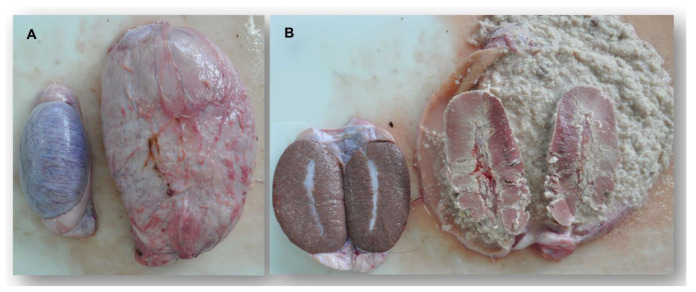
Macroscopic view of both testicles originating from a wild boar infected with *Brucella*: (**A**) the left testicle reveals a regular size and the right one is noticeably enlarged. (**B**) After dissection of the organs, the left testicle displays a physiological nature. The right testicle and epididymis are eminently enlarged and show signs of necrosis, as well as an accumulation of a large amount of purulent exudate.

**Figure 3 microorganisms-11-00478-f003:**
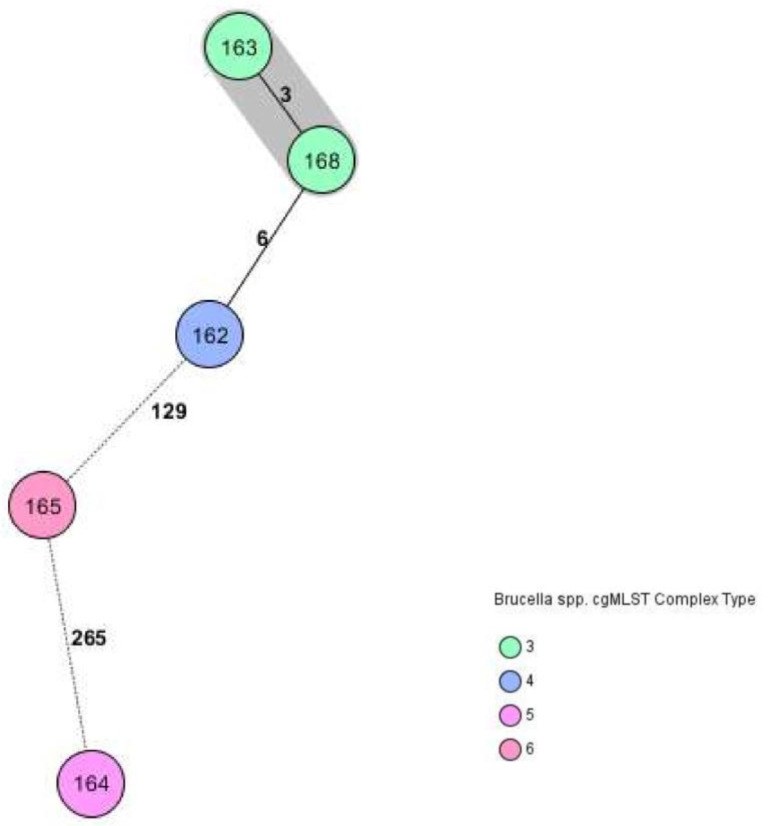
Ridom SeqSphere+ minimum spanning tree of the five Bavarian *Brucella suis* biovar 2 genomes from wild boars based on 1764 alleles. Two isolates from the same year are closely related (shaded in grey). The cluster distance threshold was 3.

**Figure 4 microorganisms-11-00478-f004:**
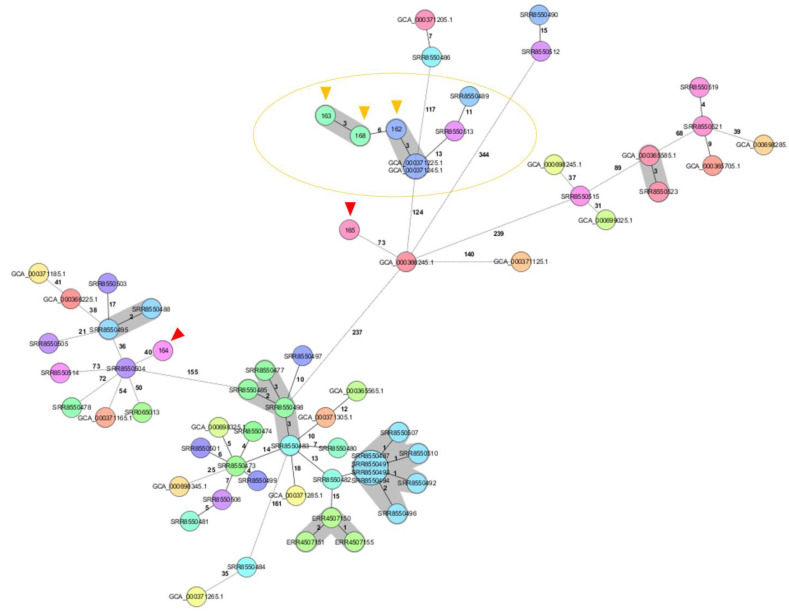
RidomSeqSphere+ minimum spanning tree including five Bavarian (red/orange arrow) and 61 other European *B. suis* genomes based on 1764 alleles and a cluster threshold of 3. The clustering isolates are shaded in grey. Different colors represent diverse lineages. Within the orange cycle, one cluster consists of two isolates, 163 and 168 (green) originating from the present study. Pictured in blue are two previously published strains from 2006, Germany (blue arrow), that cluster with the Bavarian isolate 162.

**Table 1 microorganisms-11-00478-t001:** Wild boar samples and results included in the monitoring of porcine brucellosis.

Years		2019	2020	2021	Total
Serum samples	(*n*)	3980	3899	4077	11,956
Serological reactive samples	(*n*)	716	708	717	2141
	(%)	18.0	18.2	17.6	17.9
	95%-CI	16.8–19.2	17.0–19.4	16.4–18.8	17.2–18.6
Tissue samples	(*n*)	316	160	205	681
*BCSP31*-PCR positive samples	(*n*)	2	3	3	8
	(%)				1.2
	95%-CI				0.5–2.3
Cultural isolation of *Brucella suis* biovar 2	(*n*)	1	1	3	5
	(%)				0.7
	95%-CI				0.2–1.7

*n*: number; CI: confidence interval.

**Table 2 microorganisms-11-00478-t002:** Results on the *Brucella* PCR-positive wild boars in Bavaria from 2019 to 2021.

Year/ Month	ID	Clinical Report	Sample Material	District	Administrative District	Cultural Analysis	Sequence Type	Complex Type
March 2019	162	Conspicuous clinical signs	Lymph nodes	Upper Palatinate	Amberg	*B. suis* biovar 2	16	4
April 2019	271	Found dead	Testicles	Swabia	Neuburg- Schrobenhausen	negative	n.a.	n.a.
January 2020	689	Testicles enlarged	Testicles	Upper Palatinate	Amberg- Sulzbach	negative	n.a.	n.a.
January 2020	165	Without visible symptoms	Uterus	Upper Bavaria	Eichstätt	*B. suis* biovar 2	16	6
July 2020	523	Testicles enlarged	Testicles	Middle Franconia	Nuremberg-Land	negative	n.a.	n.a.
February 2021	168	Conspicuous clinical signs	Testicles	Lower Bavaria	Kelheim	*B. suis* biovar 2	16	3
June 2021	163	Without visible symptoms	Testicles	Upper Bavaria	Freising	*B. suis* biovar 2	16	3
August 2021	164	Found dead	Testicles, Spleen	Upper Bavaria	Dachau	*B. suis* biovar 2	15	5

n.a.: not applicable.

## Data Availability

The raw sequencing data were published at the European Nucleotide Archive (ENA), project number PRJEB55696.

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
