# Peer review of "Seroprevalence of Brucella Infection in Wild Boars (Sus scrofa) of Bavaria, Germany, 2019 to 2021 and Associated Genome Analysis of Five B. suis Biovar 2 Isolates"

_microorganisms, 2023, doi:10.3390/microorganisms11020478_

Round 1

Reviewer 1 Report

Comments and suggestions

Lines 15-16 and many other parts of the ms.: (unclear) „samples“ mean the same as „individuals“? e.g., two samples of blood or organs taken from one animal are regarded as being from two animals or one animal?  Better to express it as „individuals“ or “animals“ instead of „samples“.

Fig. 1: „n“ = no. of examined animals or samples? (see above)

Line 123, Tab. 1 and other parts of the ms.: only those serum samples from one animal reactive in both ELISA plus CFT were regarded as positive – is that right?

Line 175, Tab. 2: abbreviation „n.a.“ = ?

Line 239: missing description (heads) for the pictures A and B.

Line 273: relevant studies on serosurvey of brucellosis in wild boars from other Central European countries have been omitted, but should be added. For instance: Hubálek Z., Treml F., JuÅ™icová Z., Huňady M., Halouzka J., Janík V., Bill D. (2002): Serological survey of the wild boar (Sus scrofa) for tularaemia and brucellosis in South Moravia, Czech Republic. Vet. Med. (Brno) 47: 60-66. (avg. seroprevalence rate for Brucella in wild boars was 8.7%).

Line 287 and parts discussing possible serological cross-reactions between Brucella and other bacteria (Francisella, Yersinia etc.). There is an obvious discord between a low Brucella DNA detection rate (1.2%) and the seroprevalence rate found (17.9%), despite the usually chronic brucellosis in animals results in a long-term persistence of Brucella DNA. This could really indicate a rather high proportion of false positive cross-reactions. The specificity of serological tests is here much more important than their sensitivity.

Author Response

Authors: Thank you your comments that very much helped to improve the manuscript. Please find our reworked manuscript in a newly submitted form.

Reviewer: Lines 15-16 and many other parts of the ms.: (unclear) „samples“ mean the same as „individuals“? e.g., two samples of blood or organs taken from one animal are regarded as being from two animals or one animal?  Better to express it as „individuals“ or “animals“ instead of „samples“.

Fig. 1: „n“ = no. of examined animals or samples? (see above)

Response: You are right with both comments, please see the reworked manuscript for an adequate wording.

Reviewer: Line 123, Tab. 1 and other parts of the ms.: only those serum samples from one animal reactive in both ELISA plus CFT were regarded as positive – is that right?

Response: Correct, we achieved a single result for each animal. And we rephrased the wording here and in the further text.

Reviewer: Line 175, Tab. 2: abbreviation „n.a.“ = ?

Line 239: missing description (heads) for the pictures A and B.

Response: We added the missing information for both comments.

Reviewer: Line 273: relevant studies on serosurvey of brucellosis in wild boars from other Central European countries have been omitted, but should be added. For instance: Hubálek Z., Treml F., JuÅ™icová Z., Huňady M., Halouzka J., Janík V., Bill D. (2002): Serological survey of the wild boar (Sus scrofa) for tularaemia and brucellosis in South Moravia, Czech Republic. Vet. Med. (Brno) 47: 60-66. (avg. seroprevalence rate for Brucella in wild boars was 8.7%).

Response: This is an excellent point and we included this and other references from European studies.

Reviewer: Line 287 and parts discussing possible serological cross-reactions between Brucella and other bacteria (Francisella, Yersinia etc.). There is an obvious discord between a low Brucella DNA detection rate (1.2%) and the seroprevalence rate found (17.9%), despite the usually chronic brucellosis in animals results in a long-term persistence of Brucella DNA. This could really indicate a rather high proportion of false positive cross-reactions. The specificity of serological tests is here much more important than their sensitivity.

Response: You are right. This is a topic that we discussed in more detail now in points 4.2. and 4.3. of the discussion section. Please find more details in the entire reworked discussion section.

Reviewer 2 Report

Title: It does not highlight the real content of the paper. it should be consider that the target of the  genome analysis is limited to Brucella suis biovar 2 and then rewrite the title

Abstract: It is not clear the added value and the novelties brought about by the paper. Please rewrite the abstract, including the main goals of this work and reconsider the following sentences. 

“further, a broad spreading and long term natural foci of the pathogen in Europe were confirmed” Several papers, including Munoz et al, 2019 Vet Mic 233:68-77 or older Kreizinger et al, 2014 Vet Mic 172:492-8 has already stated the endemicity of B. suis bv2 in Europe… What do the authors want to support with this sentence?

“in summary, wild boars represent a reservoir for brucellosis in Bavaria” It is not clear if this is the first report of B. suis bv2 or not; why are the authors talking about brucellosis and not swine brucellosis? Microbiology confirmed just B. suis bv2; data are lacking to define wild boar reservoir in Bavaria … serology and microbiology are applied on same samples? How to collect blood and organs?  Several older papers showed how the use of both techniques was important to define the prevalence and then the endemicity, considering the low agreement of this tests.

“However, risk of transmission to human and farm animals is still regarded low due to awareness and biosafety measures” I have some doubts about this sentence because B. suis biovar 2 (the only biovar isolated) is normally characterized by low zoonotic value and the paper does not address the issue of biosecurity measures and their effectiveness in the farms (just in pig farms or in rabbit/brown hare premises B. suis bv2 causes losses) WAHIS data reported B. suis outbreaks in 2018 2019 and 2021 both in wildlife and in pigs

Keywords: B. suis biovar 2 and genome analysis were not included .. Why not?

Introduction: Please verify the correspondence between sentece and reference. In general, as already occurred for abstract, this section is lacking of informations about the topic of the paper (B. suis, risk for humans, epidemiology, current situation in Germany and in Europe). The goals are not properly reported. Please consider to deep revise the introduction.

Line 42: I am not sure that references 6 and 7 refer to endemicity of B. suis in wildlife…please verify or find more appropriate references

Lines 47-48: where is the estimation of risk (both introduction and zoonotic) in materials/methods and results sections? Data are missing

Materials and Methods: There is a strong difference in sampling effort (blood vs organs; hunter necropsy vs laboratory examination ; genital vs limphoid tissue) not explained or supported by the authors. How was the sampling strategy set up? Which the distribution of samples among wild boar population? Please include a section or sentences describing sampling strategy and effort and limiting factors (11956 sera vs 681 tissue vs 20 whole carcasses)

Line 55-56: Do the hunters record and report to the authors lesions during the evisceration procedures?

Line 57: In other cases, which are these cases? please add useful details

Line 59: please specify a subset of animals … why just a subset?

Line 83: I think that reference is wrong. Probably the correct link is reference 9

Results: Tables and Figures need to be revised. In table 1 confidence intervals are missing. in figure 1 it is not clear where is located Bavaria Region respect on the entire Germany (not all readers know the distribution of German Region and the location can add valuable information about the epidemiology and clustering with other already positive European zone)

Line 116-122: the informations reported are the same in table 1. It’s a useless repetition.

Line 128-133: the informations reported are the same in table 1. It’s a useless repetition.

Line 151: how can you define specific lesions without histopathology? Please revise the sentence and describe the characteristics of the lesions defining suggestive of brucellar orchitis 

figure 4: the origin of the other B. suis isolates remains unknown. Please add these informations in M&M section (line 105-108)… references, database, unpublished data or what else?

Discussion: This section includes a large amount of information from reference sources and it does not deal with results obtained. Please, revise this section discussing more in depth results from seroprevalence, microbiology, gross lesions and genome analysis (totally missing

Line 249-257: these sentences are already reported as results. Please remove them from Discussion section.

Line 268-279: these sentences are, in my opinion, informations that should move to the Introduction section.

Line 278-280: please add explanation and or references for this statement

Author Response

Thank you very much for your thoughtful comments that very much helped to improve the quality of this manuscript!

Point 1: Title: It does not highlight the real content of the paper. it should be consider that the target of the  genome analysis is limited to Brucella suis biovar 2 and then rewrite the title.

Response to point 1: Right, we changed the title wording.

Point 2: Abstract: It is not clear the added value and the novelties brought about by the paper. Please rewrite the abstract, including the main goals of this work and reconsider the following sentences.

“further, a broad spreading and long term natural foci of the pathogen in Europe were confirmed” Several papers, including Munoz et al, 2019 Vet Mic 233:68-77 or older Kreizinger et al, 2014 Vet Mic 172:492-8 has already stated the endemicity of B. suis bv2 in Europe… What do the authors want to support with this sentence?

“in summary, wild boars represent a reservoir for brucellosis in Bavaria” It is not clear if this is the first report of B. suis bv2 or not; why are the authors talking about brucellosis and not swine brucellosis? Microbiology confirmed just B. suis bv2; data are lacking to define wild boar reservoir in Bavaria … serology and microbiology are applied on same samples? How to collect blood and organs?  Several older papers showed how the use of both techniques was important to define the prevalence and then the endemicity, considering the low agreement of this tests.

“However, risk of transmission to human and farm animals is still regarded low due to awareness and biosafety measures” I have some doubts about this sentence because B. suis biovar 2 (the only biovar isolated) is normally characterized by low zoonotic value and the paper does not address the issue of biosecurity measures and their effectiveness in the farms (just in pig farms or in rabbit/brown hare premises B. suis bv2 causes losses) WAHIS data reported B. suis outbreaks in 2018 2019 and 2021 both in wildlife and in pigs

Response to point 2: Thank you for these important points. Please find our rewritten abstract in the submitted new manuscript version. Also we added the suggested and other references in the manuscript below.

Point 3: Keywords: B. suis biovar 2 and genome analysis were not included .. Why not?

Response to point 3: Right, done.

Point 4: Introduction: Please verify the correspondence between sentence and reference. In general, as already occurred for abstract, this section is lacking of informations about the topic of the paper (B. suis, risk for humans, epidemiology, current situation in Germany and in Europe). The goals are not properly reported. Please consider to deep revise the introduction.

Line 42: I am not sure that references 6 and 7 refer to endemicity of B. suis in wildlife…please verify or find more appropriate references

Lines 47-48: where is the estimation of risk (both introduction and zoonotic) in materials/methods and results sections? Data are missing
Response to point 4: The introduction was deeply revised including as well the references. Please find it in the submitted manuscript. As stated above, we reconsidered the references and added more appropriate ones.

Point 5: Materials and Methods: There is a strong difference in sampling effort (blood vs organs; hunter necropsy vs laboratory examination ; genital vs limphoid tissue) not explained or supported by the authors. How was the sampling strategy set up? Which the distribution of samples among wild boar population? Please include a section or sentences describing sampling strategy and effort and limiting factors (11956 sera vs 681 tissue vs 20 whole carcasses)

Line 55-56: Do the hunters record and report to the authors lesions during the evisceration procedures? Response in lines 66-68 and in lines 127-128 (results section)

Line 57: In other cases, which are these cases? please add useful details

Response in line 64-65

Line 59: please specify a subset of animals … why just a subset?

Response in line 66-68

Line 83: I think that reference is wrong. Probably the correct link is reference 9

Response: Right, done.
Response to point 5: Please find the reworked M&M section in the submitted manuscript.

Point 6: Results: Tables and Figures need to be revised. In table 1 confidence intervals are missing. in figure 1 it is not clear where is located Bavaria Region respect on the entire Germany (not all readers know the distribution of German Region and the location can add valuable information about the epidemiology and clustering with other already positive European zone)

Response: Right, we added the confidence intervals and a map of entire Germany.

Line 116-122: the informations reported are the same in table 1. It’s a useless repetition.

Response: We skimmed this part of the text.

Line 128-133: the informations reported are the same in table 1. It’s a useless repetition.

Response: as well

Line 151: how can you define specific lesions without histopathology? Please revise the sentence and describe the characteristics of the lesions defining suggestive of brucellar orchitis

Response: Please find the rephrased text in lines 155-161.

figure 4: the origin of the other B. suis isolates remains unknown. Please add these informations in M&M section (line 105-108)… references, database, unpublished data or what else?

Response: Please find this information in lines 114-117 and Table S1.

Response to point 6: Please also find the reworked results section in the newly submitted manuscript.

Point 7: Discussion: This section includes a large amount of information from reference sources and it does not deal with results obtained. Please, revise this section discussing more in depth results from seroprevalence, microbiology, gross lesions and genome analysis (totally missing

Response: Right, we included a new paragraph 4.4 now.

Line 249-257: these sentences are already reported as results. Please remove them from Discussion section.

Response: Right, we strongly skimmed this part.

Line 268-279: these sentences are, in my opinion, informations that should move to the Introduction section.

Response: Right, we moved these up there.

Line 278-280: please add explanation and or references for this statement

Response: Right, we rephrased this part of the discussion. It is now paragraph 4.2.

Response to point 7: Please also find the reworked discussion section in the submitted manuscript.

Reviewer 3 Report

I read the manuscript presented to me for review with great attention. The results described by the authors are very interesting. I have no comments on the research methodology, which is in line with OIE standards. My only remark to the authors was the suggestion that they should include the results of research from the neighboring country of Germany, Poland, where research in this area was also conducted. I suggest to add the situation on wild boars in Poland where also Brucella suis 2 was isolated. I rate the manuscript very highly and have no other comments to make. My assessment results from the fact that although the results are presented correctly, the authors themselves have noticed that Europe is brucellosis-free and the risk of transmission to humans is negligible.

Do you consider the topic original or relevant in the field? Does it address a specific gap in the field? YES

• What does it add to the subject area compared with other published material? Wild-life animals are reservoir for Brucellosis and they should be considered as low risk potential pathogenic for humans or domestic animals.

• What specific improvements should the authors consider regarding the methodology? What further controls should be considered? Methodology is correct.

• Are the conclusions consistent with the evidence and arguments presented and do they address the main question posed? The conclusions are good.

• Are the references appropriate? YES

Author Response

Thank you very much for your comments that helped to improve the quality of our manuscript! Please find our changes in the attached letter and the revised manuscript.

Reviewer 4 Report

The manuscript by LM Luaces et al. entitled "Prevalence and Genome Analysis of Brucella in Wild Boars (Sus scrofa) in Bavaria, Germany, 2019 to 2021" is devoted to the isolation and characterization of Brucella isolated from wild boars in Bavaria. The authors performed WGS on five strains of B. bovis biovar 2. The manuscript is well done, but I would recommend an English editing.

Other minor issues are listed below:

Lines 348 and 387: what is meant by "here"?

Line 540, Table 1, line 661: italicize bcsp31

2.4: I suggest that the project number of the deposited sequence data should be given here, according to the Instructions for Authors: All sequence names and accession numbers provided by databases must be provided in the Materials and Methods section of the article.

Line 744: Again, move back to the "Materials and Methods" section.

Lines 790 and 817: Change "columns" to "alleles".

Figure 4: Why repeat part of 4A in 4B? Just circle it in the square in 4A.

Line 1458: Add the accession numbers / BioProject number here.

Author Response

Thank you very much for your comments that helped to improve the quality of our manuscript!

Please find our answers to your comments in the attached letter and the reworked manuscript.

Reviewer 5 Report

Dear Authors, I am of the opinion that you have presented an interesting and highly topical study, also considering the European problem of the spread and overpopulation of wild boars, and the related risks to public health.

There are some changes to be made which I indicate below as suggestions and observations:

1. Line 25 - Keywords: I think some of the keywords listed are already in the title, so it's neither necessary nor helpful to repeat them in this section. Please remove "seroprevalence", "genome analysis", "Bavaria" and "Germany"

2. Lines 56-70 - The "Materials and Methods" section must be modified and above all, some aspects related to sampling must be clarified. a) Was the evisceration of wild boars after killing done without the presence of a Veterinarian? b) how was the serum or transudate collected?

3. Line 82 - Please change "autopsy" to "necropsy," which is more appropriate for postmortem examination performed on animals.

4. Line 128 - I apologize if I insist on this aspect, but only a Veterinarian performs the clinical evaluation. It would be appropriate to rephrase this sentence correctly.

5. Lines 259-343 - If the Authors do not deem it essential, I suggest avoiding dividing the Discussion into subsections. Please modify. Furthermore, I suggest inserting the "Conclusions" section, even if it is not mandatory, or alternatively adding a final short paragraph in the Discussion to conclude the paper in a manner consistent with the title and the results.

Author Response

Thank you very much for your review and critical comments that helped to improve the quality of our manuscript!

Please find the response in the attached pdf and the resubmitted manuscript.

Reviewer 6 Report

I believe that the Authors revised the manuscript in a manner consistent with the requirements of the referees. In this new form the manuscript can without any further delay be published.

Author Response

Thank you very much for your evaluation!

Round 2

Reviewer 2 Report

Reply to v1 version (right new title)

Abstract: "In summary, wild boars represent a reservoir for brucellosis in Bavaria. However, risk of transmission to humans and farm animals is still regarded low due to awareness and biosafety measures." Please specify swine brucellosis or brucellosis due to B. suis

Discussion:

Lines 249-252: remove it. This data are results not discussion. Please insert a brief introducing sentence (i.e. consider to report lines 311-328 or some parts of them)

Lines 258-258: move to results (paragraph 3.1)

Lines 315-316: please verify the correct use of at this time (when? now or 1980's), or rewrite the sentence.

Lines 329-333. please consider to insert first the two last sentences and then conclude with "Therefore, swine brucellosis should not be underestimated as a rare but serious pathogen for different animal species and as a zoonosis."

Author Response

Thank you very much for your repeat review and critical comments that helped to improve the quality of our manuscript!

Please find the response in the attached pdf and the resubmitted manuscript.
